# A Combination of Full Pulpotomy and Chairside CAD/CAM Endocrown to Treat Teeth with Deep Carious Lesions and Pulpitis in a Single Session: A Preliminary Study

**DOI:** 10.3390/ijerph17176340

**Published:** 2020-08-31

**Authors:** Marie-Laure Munoz-Sanchez, Natacha Linas, Nicolas Decerle, Emmanuel Nicolas, Martine Hennequin, Pierre-Yves Cousson

**Affiliations:** 1Centre de Recherche en Odontologie Clinique (CROC), Université Clermont Auvergne, F-63000 Clermont-Ferrand, France; mlmunozsanchez@gmail.com (M.-L.M.-S.); natacha.linas@uca.fr (N.L.); nicolas.decerle@uca.fr (N.D.); emmanuel.nicolas@uca.fr (E.N.); p-yves.cousson@uca.fr (P.-Y.C.); 2CHU Clermont-Ferrand, Service d’Odontologie, F-63003 Clermont-Ferrand, France

**Keywords:** deep carious lesions, pulpitis, full pulpotomy, chairside, CAD/CAM restoration

## Abstract

A higher chance of carrying out a successful full pulpotomy may depend on whether the coronal restoration can be completed within a single appointment. The development of chairside CAD/CAM (Computer Aided Design and Manufacturing) technology has made it possible to carry out indirect restoration of endodontically treated teeth in a single session. This study aimed to evaluate the long-term outcome of a full pulpotomy with Biodentine™ immediately covered with a chairside CAD/CAM endocrown on teeth affected by pulpitis and deep carious lesions. The investigation involved a cohort of 30 molars that were treated by pulpotomy and CAD/CAM endocrown. Clinical and radiological examinations were scheduled at 1, 6, and 12 months postoperatively. Overall, all treatments were effective at any time during the follow-up. The results of this study need to be confirmed with a longer-term follow-up to allow for comparison with the literature. This original combination of endodontic and restorative treatments provides an Endo-prosthetic continuum in a single session, with the objective of long-term success in terms of tooth health.

## 1. Introduction

Over the last decade, aided by simultaneous progress in biological research and development of materials, recommendations have been made for carious tissue removal and caries management in vital teeth with the aim of preserving tooth tissue and retaining teeth in the long term. Minimum intervention principles now guide both restorative dentistry and endodontics [1,2]. When a tooth with severe coronal tissue damage is diagnosed with reversible pulpitis, there are two possible minimal intervention approaches. The conservative approach aims to prevent pulpal exposure and to induce pulpal reaction by the formation of reparative dentine in the periphery of the cameral pulp, localized at the front of the carious decay. This approach aims to maintain tissue integrity by treating reversible pulpitis with a stepwise removal technique and then restoring the loss of coronal tissue with plastic materials. The success rate for this procedure varies widely from 56% to 100% [3,4,5]. The alternative approach is to treat pulpitis with a full pulpotomy (removal of the entire coronal pulp to the level of the canal) [6,7,8,9] to restore coronal damage with direct plastic materials with cusp coverage [9] or preformed crowns [10]. This approach is more invasive than the first option, but can be seen as a “preservative approach” as its goal is to preserve the tooth on the arch in the long term. The success rate of full pulpotomy procedures varies between 82.9% and 100%, depending on the coronal restoration [8,9,11,12,13]. A better tooth survival rate is found when the restoration is carried out within two days of pulp exposure, highlighting the need to limit the number of treatment sessions as well as the time between successive sessions [14]. Thus, as for all vital pulp therapy, the degree of success for full pulpotomy may depend on whether the coronal restoration can be carried out within the same appointment. 

For deep carious lesions or teeth weakened by considerable cavity preparation, indirectly bonded restoration is more suitable than direct restoration [15]. Survival rates of teeth with carious cavities on more than three surfaces are higher when restored with full or partial crowns rather than with composite restorations [16]. Teeth that are indirectly restored with composite or ceramic have better fracture resistance and marginal integrity, reduced cervical marginal microleakage and less surface roughness, postoperative sensitivity, and soft-tissue irritation than those directly restored with composite [15,17,18,19,20]. Overall, indirect restorations have a lower annual mean failure rate than direct restorations in posterior teeth [21]. 

There may be a benefit in restoring pulpotomised teeth with deep carious lesions using indirect restorations. The development of chairside CAD/CAM (Computer Aided Design and Manufacturing) technology has made it possible to carry out indirect restoration of a tooth in one single session. This study aimed at evaluating the outcome of the original combination of full Pulpotomy and Immediate CAD/CAM Endocrown (PICCE) on teeth with large carious lesions and pulpitis.

## 2. Materials and Methods 

### 2.1. Type of Study

This is a case report on a series of patients based on data collected during dental care sessions. Study Ethics approval was obtained on 15 may 2020 (CECIC Rhône-Alpes-Auvergne, Grenoble, IRB 5921). The study was conducted in accordance with the Declaration of Helsinki. All patients were given information about the study and gave their consent to participate.

### 2.2. Patients 

Patients presenting one molar with a deep or extremely deep carious lesion, according to Bjørndal et al. [22], associated with a diagnosis of reversible, chronic or irreversible pulpitis were recruited from the Dental Department of the University Hospital of Clermont-Ferrand between November 2017 and January 2020. The full list of inclusion and exclusion criteria is presented in Table 1.

Preoperative pulpal and periapical diagnoses were established after clinical and radiological examination. Preoperative retro-alveolar radiographs were taken using film holders and a paralleling technique.

### 2.3. Pulpotomy and Restoration Procedures

Pulpotomy and tooth restoration were conducted chairside over the course of a single session. After local or locoregional anesthesia (with adrenaline solution at 1:200,000), the rubber dam was placed. The pulp chamber was opened using a sterile high-speed diamond bur under water coolant; the access cavity was created with 21 mm Endo-Z bur. From this point on, abundant irrigation with a 2.5% sodium hypochlorite solution was performed. Pulpotomy is performed until canal orifices level, unless hemostasis is not reached. In that case, the pulp tissue was amputated at 2 mm beneath the level of the canal orifices using a Gates drill. Hemostasis was achieved by applying a cotton pellet moistened with 2.5% sodium hypochlorite solution for 3 min and repeating, if required, for up to 6 min. Biodentine™ was mixed according to the manufacturer’s instructions and placed in a 3-mm or 4-mm layer above the pulp tissue. After the initial setting time of 12 min, the rubber dam was removed for coronal preparation. 

An occlusal clearance of at least 2 mm was arranged under the entire occlusal surface in order to save space for the ceramic restoration with an objective of full cuspal coverage. As opposed to the endocrown preparations described for root-filled teeth, the central anchorage of the pulpotomised tooth’s endocrown was sited within the pulp chamber, maintaining the Biodentine™ thickness to about 3 mm over the pulpal floor and eliminating undercuts in the access cavity (Figure 1). 

The preparation limits were supragingival to facilitate impression and bonding. An optical impression was obtained with a Cerec^®^ Omnicam camera (Figure 2A–D). The rubber dam was then replaced. The endocrowns were designed and manufactured in blocks of IPS e.max^®^ CAD or Enamic^®^ depending on the clinical case (Figure 2E,F). Manufacturing time was about 15 min for both materials. IPS e.max^®^ CAD endocrowns were tried on the teeth before being fired for 25 min. All endocrowns were etched with hydrofluoric acid for 20 s for IPS e.max^®^ CAD and 30 s for Enamic^®^ respectively. After priming, endocrowns were sealed with a dual cure resin cement (Variolink^®^ Esthetic DC) (Figure 3). A postoperative periapical radiograph was taken after restoration placement using a film holder and a paralleling technique.

### 2.4. Follow up Evaluations 

Patients were asked to come back for a check-up 1 month (T1), 6 months (T2), and one year (T3) after treatment. At each stage, a clinical and radiographic examination of the treated tooth was carried out (Figure 4). Clinical examination was conducted to verify: (i) the presence of the tooth on the arch; (ii) the lack of dental pain or pain-related behavior declared by the patient, and (iii) the lack of clinical symptoms of infectious disease related to the therapeutic pulpotomy treatment (oedema, fistula or tooth mobility). Retro-alveolar radiographs were taken using film holders and a paralleling technique. Radiological evaluation was conducted to evaluate and compare the Periapical Index (PAI) score of the treated tooth after the follow-up period, with the initial PAI score being T0 [23]. Pulp canal calcification was also radiologically checked throughout the follow-up period.

### 2.5. Study Criteria

The outcome of the pulpotomy was evaluated on the basis of clinical and radiological criteria (Table 2) [24]. 

Two investigators were trained to interpret PAI scores with a test and retest 15 days apart on 100 X-rays illustrating the five score categories. The intra-class correlation coefficient (ICC) for inter- rater assessment was 0.95 (*p* < 0.001) for the test phase and 0.93 (*p* < 0.001) for the retest at 15 days. Intra-rater validity was 0.87 (*p* < 0.001) for the first expert and 0.90 (*p* < 0.001) for the second expert. The first examiner’s reliability was 0.87 (*p* < 0.001) relative to the expert panel on the test and 0.89 (*p* < 0.001) on the retest at 15 days, while the reliability relative to the expert panel for the second examiner was 0.86 (*p* < 0.001) on the test and 0.88 (*p* < 0.001) on the retest. 

For the evaluation of radiological criteria, all postoperative images were proposed in a random order and interpreted by two calibrated investigators. In the event of a disagreement, a consensual decision was reached between both readers and a third calibrated investigator.

### 2.6. Statistical Analysis

Descriptive analysis was carried out in terms of the percentage of teeth satisfying the success criteria. BiostaTGV (Sentinelles Network, Paris, France) was used to calculate the number of inclusions necessary to achieve a success rate for teeth treated with Biodentine™ pulpotomy and immediately restored that equals or exceeds that for vital teeth treated by the stepwise excavation technique (non-inferiority hypothesis) [3]. The number of required subjects observed one year after treatment was 8 (α = 5%, β = 10%).

## 3. Results

Thirty patients were included in this study, with a total of 16 lower and 14 upper molars (Table 3). 

Eight molars were examined at each follow-up stage of the study. The recruitment of the required number of subjects took 27 months. 

The flow chart of the participants in the cohort is presented in Figure 5. 

Twelve patients (40%) were lost to the study during follow-up. The distribution of the teeth according to the clinical and radiological study criteria is reported in Table 4. 

Overall, no ineffective pulpotomies were observed regardless of the follow-up duration. After one and six months, 23/28 (82%) and 15/16 (94%) of observed pulpotomies, respectively, were effective. All pulpotomies evaluated after one year (8/8) were effective, and with no canal calcification. In most of the cases, the categorization into uncertain pulpotomies was related to clinical, rather than radiological, criteria.

## 4. Discussion

This is the first study to describe the outcome of immediate indirect restorations for pulpotomised teeth. The analysis of this study was based on a non-inferiority hypothesis, which assumed that long-term outcomes of PICCE would be at least equivalent to those of a stepwise technique followed by a direct permanent or temporary restoration. This hypothesis was verified. 

This study presents some limitations related especially to the analysis strategy and high rate of subjects lost to follow up. Indeed, the data were not analyzed using an intention-to-treat approach. An analysis carried out whereby the denominator (*n* = 30) stays the same regardless of what happens to the patient, the assumption made is that all the patients who were lost to follow up had negative outcomes, then the success rate would be significantly less favorable to the treatment procedure presented. As such, it is likely that the true success rate lies somewhere between the percentages given and the percentages obtained using an intention-to-treat analytical approach. The difficulty of strictly following the study protocol within the ethical setting and consequently the high loss to follow up rate, impacts the results. In addition, the study design does not include a randomization procedure and a control group. Consequently, the results of this study need to be confirmed in a longer-term follow-up to allow comparison with data in the literature, as the indirect CAD/CAM endocrowns are expected to provide better long-term results than direct restorations in pulpotomised teeth. Discussing these results within the context of preservation/conservation provides a new perspective to justify this minimalist dentistry approach.

Failed pulp therapy could be caused by one of two main aetiologies, the first being related to the pulp’s inability to respond to inflammation or infection, and the second being related to insufficient sealing between the pulp capping material and the coronal restoration. The causes for pulpal aetiologies could be due to either extended inflammation resulting from an imprecise diagnosis or peroperative contamination. In such situations, treatment failure occurs in the weeks directly following the procedure. On the other hand, failed pulpotomies due to coronal contamination are delayed, occurring several months or years after treatment. 

There is still debate over the use of full pulpotomy as a treatment for pulpal disease, as this depends on the pulpal status. In previous studies evaluating the outcome of full pulpotomy in permanent teeth, the terminology used to describe the teeth concerned varied considerably. Different terms, such as reversible pulpitis, irreversible pulpitis, symptomatic teeth, vital teeth with carious exposure, and chronic pulpitis, have been used [8]. The terminology for clinical pulp diagnosis and pulpal treatment conditions is currently being debated [2,25]. The variability of the diagnostic terminology raises the question as to the pertinence of categorizing all these pulpal statuses as different diseases. It has already been demonstrated that the histological state of the pulp is not related to the clinical signs and symptoms [26,27,28]. Consequently, the pulpal diagnosis is based on clinical and radiological signs. Moreover, while endodontists consider endodontic treatment to be fully complete after coronal restoration, the status of pulpal and coronal decay is not taken into account when establishing the diagnosis. Returning to the definition of a disease could help with understanding the clinical issues around this debate. According to the historical medical definition, a disease is characterized by an aetiology, a pathogeny, a semiological context (signs and symptoms) and a treatment, which is specific to the given disease. This study suggests that the association of pulpal inflammation and deep carious lesion would constitute an aetiological entity for a disease for which the combination of full pulpotomy and immediate placement of a CAD/CAM endocrown would be the specific treatment. In this case, the term pulpal inflammation groups together different inflammatory statuses, including reversible, irreversible or chronic pulpitis, all of which seem to be successfully treated by PICCE. PICCE could therefore be defined as the preservative approach for a tooth with pulpitis and deep carious lesion. 

Recent trials and studies [6,9,29] which selected vital pulp and carefully avoided per-operative contamination have been carried out successfully regardless of pulpal diagnosis. Thus, it could be hypothesized that the reported failure rates were due to post-operative coronal contamination rather than pulpal aetiologies. In cases involving deep carious lesions, residual dentine would be more or less demineralized, which would possibly reduce the mechanical performance of bonded restorations [30,31]. Moreover, the greater extent of coronal tissue loss necessitates bigger scale restorations, with cusp and margin reconstitutions, which could increase the risk of fracturing and partial or total debonding of direct restorations. When teeth are treated under a two-visit protocol separated by an interval of 8–12 weeks, there is a risk that the temporary cement will compromise the coronal seal, particularly for cavities with fewer than four residual walls. These complications lead to bacterial contamination, impeding the healing potential of the pulp. This healing ability is already affected by the carious process, in addition to the conservative procedure itself. A stepwise, two-stage removal could mean that the fibroblast batch is consumed prematurely, which has a further negative effect on the restorative response of the pulp in the face of new aggressions.

In most teeth with pulpitis, the pulp chamber dentine is sound, providing an excellent substrate for optimal adhesion. The arrangement of the cameral cavity increases the surface of sound dentine available for bonding indirect restorations [32,33,34]. Biodentine™ allows a definitive restoration to be bonded immediately after its initial setting time [35,36,37]. Endocrowns are restorations guided by conservative principles [34,38]. Compared to inlays, endocrowns appear to be an effective solution for restoring severely damaged posterior teeth because they provide potential protection against debonding at the dentine-restoration interface and increased crown stiffness [39].

## 5. Conclusions

Within the limitations of this preliminary study, full Pulpotomy and Immediate CAD/CAM Endocrown (PICCE) could be safely recommended to treat, during a single session, severely damaged permanent molars with pulpitis. This new combination preserves the residual biological potential of the pulp, ensuring an endo-prosthetic continuum with the objective of long-term success.

## Figures and Tables

**Figure 1 ijerph-17-06340-f001:**
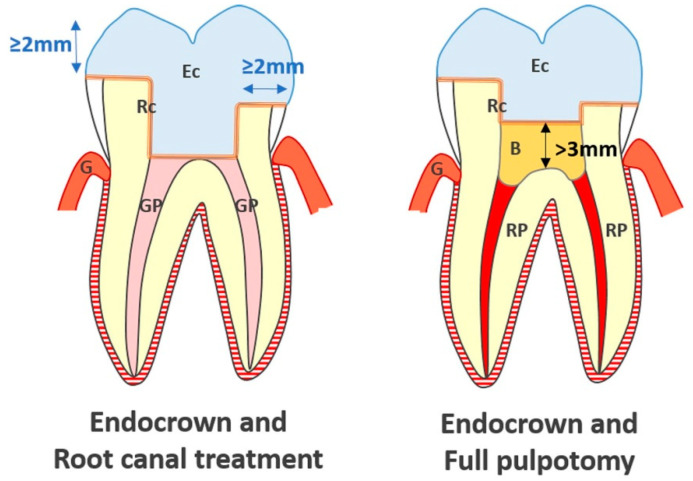
Schematic comparison of central anchorages of endocrowns carried out on teeth treated with root canal treatment or full pulpotomy (Ec: Endocrown; Rc: Resin Cement; GP: Gutta Percha; G: Gingiva; B: Biodentine™; RP: Radicular Pulp).

**Figure 2 ijerph-17-06340-f002:**
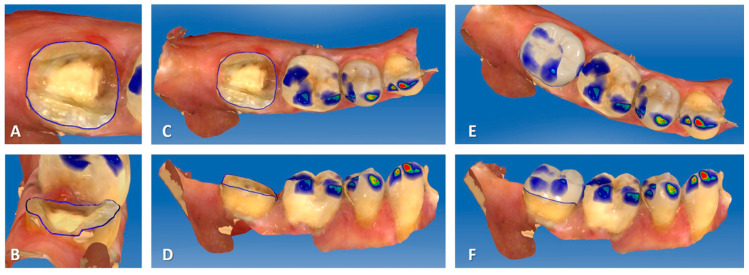
Cerec^®^ screenshots for endocrown modeling ((**A**): occlusal view of the preparation; (**B**): distal view; (**C**): occlusal view of the quadrant; (**D**): buccal view of the quadrant; (**E**): occlusal view of the Cerec^®^ endocrown model before machining; (**F**): buccal view of the Cerec^®^ endocrown model before machining).

**Figure 3 ijerph-17-06340-f003:**
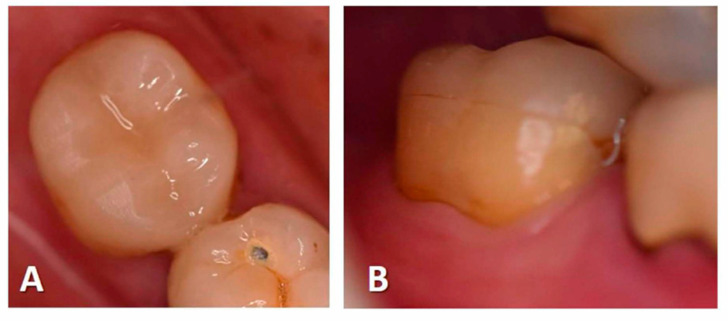
Intraoral photographs of tooth 47 at 18 months postoperative ((**A**): occlusal view; (**B**): buccal view).

**Figure 4 ijerph-17-06340-f004:**
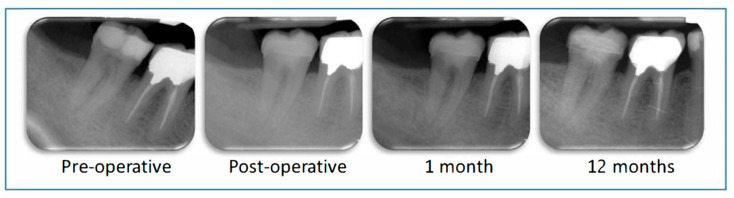
Radiological monitoring of tooth 47 during one year post-operatively. The Periapical Index (PAI) score remains at 1 throughout the follow-up period.

**Figure 5 ijerph-17-06340-f005:**
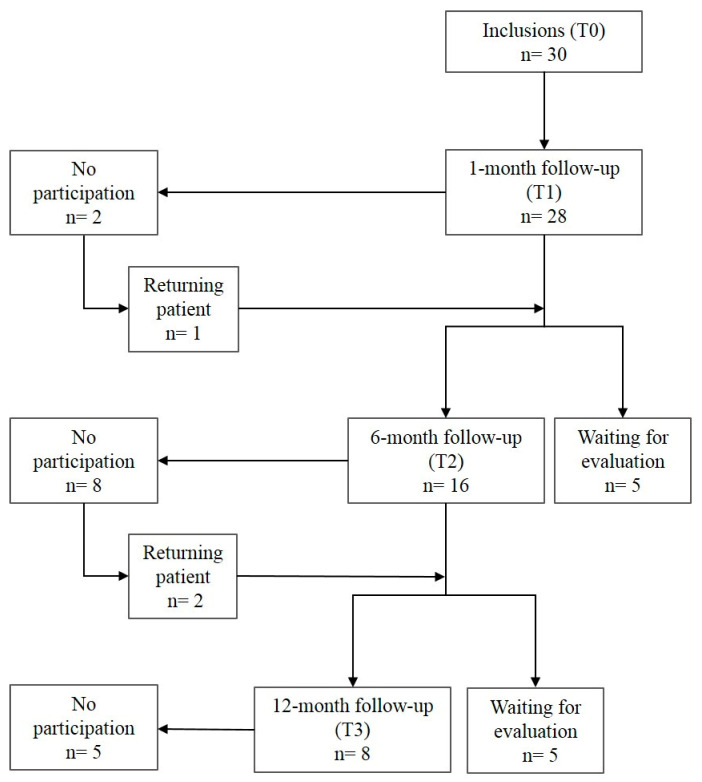
Flow chart of the study.

**Table 1 ijerph-17-06340-t001:** Inclusion and exclusion criteria.

Inclusion criteria	- ***Related to the patient*** ○ Patient accepting the protocol, speaking French, without contributory medical history
	- ***Related to the tooth*** ○ Mature permanent molar ○ Possibility of installing a rubber dam ○ Number, thickness and location in relation to the gingival limit of residual walls remaining after curettage compatible with a CAD/CAM restoration ○ Absence of periodontal lesion ○ Restorable teeth, probing pocket depth and mobility are within normal limits
	- ***Related to the pulpal status*** ○ Clinical diagnosis of reversible (low intense, short-lasting induced pain, positive response to vitality tests, without pain on apical palpation of the soft tissues or percussion pain, no radiologically visible apical image, PAI 1 or 2) or irreversible (spontaneous, radiating pain that lingers after removal of the stimulus, without pain on apical palpation of the soft tissues or percussion pain, no radiologically visible apical image, PAI 1 or 2) pulpitis ○ Clinical diagnosis of chronic irreversible (episodes of spontaneous or induced pain, positive response to vitality tests, without pain on apical palpation of the soft tissues or percussion pain, no radiologically visible apical image, PAI 1 or 2) pulpitis
Exclusion criteria	- ***Related to the patient*** ○ Patient refusing the protocol, not understanding French, with medical contraindication
	- ***Related to the tooth*** ○ Temporary teeth, immature permanent teeth ○ Number, thickness and location in relation to the gingival limit of residual walls remaining after curettage compatible with a direct restoration (amalgam or resin composite) ○ Non restorable teeth ○ Impossibility to install a rubber dam ○ Periodontal pathology ○ No pulp exposure after caries excavation
	- ***Related to the pulpal status*** ○ Necrotic and/or infected tooth (negative response to the vitality tests, pain on apical palpation of the soft tissues, percussion pain, radiologically visible apical image, PAI 3, 4 or 5)
	- ***Extemporaneous exclusion criteria*** ○ Identification during the protocol of necrosis of at least one of the root canals (absence of bleeding): partial or total necrosis ○ Bleeding could not be controlled after full pulpotomy in 6 min

**Table 2 ijerph-17-06340-t002:** Zanini et al. criteria for the evaluation of the outcome of pulpotomy.

Outcome of Pulpotomy	Clinical Criteria		Radiographic Criteria
	Functional Tooth	Noninfected Tooth	
Success, effective pulpotomy	Lack of pain declaration and Presence of the tooth and Sealing properties of the restoration	Absence of spontaneous pain and Absence of pain on chewing and Lack of swelling and Lack of swelling and sinus tract and Negative response to axial percussion test and Negative response to apical palpation test and Periodontal probing <2mm	PAI at T0 = 1 and PAI at Tx = 1, PAI at T0 = 2 and PAI at Tx ≤ 2, or PAI at T0 ≥ 3 and PAI at Tx ≤ 2 and lack of radicular lacunae
Uncertain outcome	Lack of pain declaration and Presence of the tooth and Sealing properties of the restoration	Absence of spontaneous pain and Absence of pain on chewing and Lack of swelling and Lack of swelling and sinus tract and Negative response to axial percussion test and Negative response to apical palpation test and Periodontal probing <2mm	PAI at T0 = 1 and PAI at Tx = 2, PAI = 3 at both T0 and Tx, and lack of radicular lacunae
Failure, ineffective pulpotomy	Lack of pain declaration and/or Presence of the tooth and/or Sealing properties of the restoration	Absence of spontaneous pain and/or Absence of pain on chewing and/or Lack of swelling and Lack of swelling and sinus tract and/or Negative response to axial percussion test and/or Negative response to apical palpation test and/or Periodontal probing <2mm	PAI at T0 = 1 or 2 and PAI at Tx ≥ 3, PAI at T0 ≥ 3 and PAI at Tx > 3 and/or presence of radicular lacunae

PAI, Periapical Index; T0: date of treatment; Tx: longest follow-up times (T1, T2 or T3).

**Table 3 ijerph-17-06340-t003:** Descriptive criteria for the included teeth at the initial (T0) and final evaluation (T3).

Descriptive Criteria	Initial Evaluation (T0)	12 Month Follow-Up (T3)
**Diagnosis**		
Reversible pulpitis	23	5
Irreversible pulpitis	2	1
Chronic pulpitis	5	2
Total	**30**	**8**
**Bjørndal classification**		
Deep carious lesion	14	4
Extremely deep carious lesion	14	4
No carious lesion	2	
Total	**30**	**8**
**Endocrown material**		
IPS e.max^®^ CAD	15	2
Enamic^®^	15	6
Total	**30**	**8**

Total numbers of teeth in each category were noted in bold.

**Table 4 ijerph-17-06340-t004:** Distribution of teeth according to the outcome of the combined treatment (full pulpotomy and immediate coverage with endocrown) at different follow-up times: T1 (1 month), T2 (6 months) or T3 (12 months).

Outcome	Follow-up Time	Clinical Criteria		Radiographic Criteria	Total
		Functional Tooth	Non-Infected Tooth		
Effective outcome	T1	28 (100%)	24 (86%)	27 (96%)	**23**
	T2	16 (100%)	16 (100%)	15 (94%)	**15**
	T3	8 (100%)	8 (100%)	8 (100%)	**8**
Uncertain	T1	0	4 (14%) *	1 (4%)	**5**
	T2	0	0	1 (6%)	**1**
	T3	0	0	0	**0**
Ineffective outcome	T1	0	0	0	**0**
	T2	0	0	0	**0**
	T3	0	0	0	**0**

* all four cases are related to percussion pain; Total numbers of teeth in each category of outcome were noted in bold.

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
