# Peer review of "A Combination of Full Pulpotomy and Chairside CAD/CAM Endocrown to Treat Teeth with Deep Carious Lesions and Pulpitis in a Single Session: A Preliminary Study"

_ijerph, 2020, doi:10.3390/ijerph17176340_

Round 1
Reviewer 1 Report
Dear Authors
I must commend all of you for a very well written paper. The concept is good and the layout and presentation of the tables, etc were excellent. I do have some areas of concerns where I think you can improve the content and quality of the paper.
- Study Design : This is basically a case report on a series of patients who presented with deep carious lesions in their permanent molars and were treated with a combination of full pulpotomy and chair-side CAD/CAM endocrown in a single session.
- Results: Whilst the results are well presented, my view is that you have been overly optismistic about the success of your treatment protocol simply because you have not analysed the data using an intention to treat approach whereby the denominator (n=30) stays the same regardless of what happens to the patient. So whilst you have reported vey high success rates at one month (23/28; 82%); 6 months (15/16;94%) and one year (8/8; 100%), your loss to follow up rate of 40% is extremely high. Using an intention to treat approach, on has the following success rates of 23/30 (77%); 15/30 (50%) and 8/30 (27%) at 1, 6 and 12months respectively. The assumption made is that all the patients who were lost to follow up had negative outcomes. so, in reality, the true effective (success rates) lies somewhere between your reported percentages and the percentages obtained using an intention to treat analytical approach. To be fair to the reader, I think it is extremely important that you discuss the impact of the high loss to follow up rate on the interpretation of your results. I think you need to include this in the discussion section under limitations of your study. You may use the above if you so wish. You may also wish to highlight the limitation of your study design.
- I think your conclusion is written in an overly optimistic manner. Your opening sentence should contain the words "Within the limitations of this study......."
- Again this is a very well written study but I do think that the authors have been overly optimistic with their results.
Author Response
- Study Design: This is basically a case report on a series of patients who presented with deep carious lesions in their permanent molars and were treated with a combination of full pulpotomy and chair-side CAD/CAM endocrown in a single session.
Answers: It is indeed a descriptive study that tracks the success rate of this treatment procedure.
- Results: Whilst the results are well presented, my view is that you have been overly optismistic about the success of your treatment protocol simply because you have not analysed the data using an intention to treat approach whereby the denominator (n=30) stays the same regardless of what happens to the patient. So whilst you have reported vey high success rates at one month (23/28; 82%); 6 months (15/16;94%) and one year (8/8; 100%), your loss to follow up rate of 40% is extremely high. Using an intention to treat approach, on has the following success rates of 23/30 (77%); 15/30 (50%) and 8/30 (27%) at 1, 6 and 12months respectively. The assumption made is that all the patients who were lost to follow up had negative outcomes. so, in reality, the true effective (success rates) lies somewhere between your reported percentages and the percentages obtained using an intention to treat analytical approach. To be fair to the reader, I think it is extremely important that you discuss the impact of the high loss to follow up rate on the interpretation of your results. I think you need to include this in the discussion section under limitations of your study. You may use the above if you so wish. You may also wish to highlight the limitation of your study design.
Answers:
We are in full agreement with this point of view, we thank you for this suggestion and propose to include the following elements in the discussion:
“This study presents some limitations especially related to the analysis strategy and high rate of subjects lost to follow up rate. Indeed, the data was not analyzed using an intention to treat approach. An analysis carried out whereby the denominator (n=30) stays the same, regardless of what happens to the patient, the assumption made is that all the patients who were lost to follow up had negative outcomes, the success rate would be significantly less favorable to the treatment procedure presented. As such, it is likely that the true success rate lies somewhere between the percentages given and the percentages obtained using an intention-to-treat analytical approach. The difficulty of strictly following the study protocol within the ethical setting and consequently the high loss to follow up rate, impacts this study results. In addition, the study design does not include a randomization procedure and a control group. Consequently, the results of this study need to be confirmed in a longer-term follow-up to allow comparison with data in the literature, as the indirect CAD/CAM endocrowns are expected to provide better long-term results than direct restorations in pulpotomised teeth.”
- I think your conclusion is written in an overly optimistic manner. Your opening sentence should contain the words "Within the limitations of this study......."
Answers: we qualify the conclusion as follows:
“Within the limitations of this preliminary study, full Pulpotomy and Immediate CAD/CAM End-crown (PICCE) could be safely recommended to treat, during a single session, severely damaged permanent molars with pulpitis. This new combination preserves the residual biological potential of the pulp, ensuring an endo-prosthetic continuum with the objective of long-term success.”
- Again this is a very well written study but I do think that the authors have been overly optimistic with their results.
Answers: We would like to thank Reviewer1 for his/her help, so we can qualify our results.
Reviewer 2 Report
The manuscript titled “A combination of full pulpotomy and chairside CAD/CAM end-crown to treat teeth with deep carious lesions and pulpitis in a single session: a preliminary study” aims to evaluate the effect of chairside CAD/CAM end-crown in human clinical trial.
Introduction
Pulpotomy includes partial pulpotomy, cervical pulpotomy full pulpotomy…etc. The word full pulpotomy is used in this manuscript. Add the explanation pulpotomy and define.
“This study aimed at evaluating the outcome of the original combination of full Pulpotomy and Immediate CAD/CAM End-crown (PICCE) on teeth with large carious lesions and pulpitis”(Page 2, Line 67). Did the conclusion correspond to individual sentences? The author investigated success rate of CAD/CAM endo crown without control. This should be stated as limitation.
Material and methods
The author described “All pieces were polished before bonding. End-crowns were sealed with a dual cure resin cement” (Page 4, Line 116). Bonding procedure is important for CAD/CAM restoration. Did you perform sandblasting and priming??
Describe details such as clinical procedures. Microscope video should be shown as new figure. Presenting these results proves your procedure.
In addition, the author described “A postoperative periapical radiograph was taken after restoration placement using a film holder and a paralleling technique.” (Page 4, Line, 118). Clinical date such as radiographs are not enough in this manuscript. Representative series of clinical images (X-ray photo, CBCT image) and videos (microscope), intraoral photo of each group should be open to help the reader. Especially, pre-ope, post-ope and follow-up, these time points is important.
Result
In figure1 End-crown and full pulpotomy, where is circular butt joint??
In addition, photograph of end-crown made using CAD/CAM is should be added as figures.
“12 patients (40%) were lost to the study during follow-up.“(Page 8, Line163).
It seems high rate. Is this acceptable??
Additionally, how did you control the bias through this study?.
Discussion
The author evaluated shor time out come of CAD/CAM endocrown, but did’t mention long trem follow-up. This part sould be stated as limitation of this paper.
Result of this study should be compared with previous study. Additionaly, this study was not randamized control trial. It is important to note limitation.
Conclusion
“This new combination preserves the residual biological potential of the pulp, ensuring an endo-prosthetic continuum with the objective of long-term success.” ((Page 8, Line 235). Folow-up period was one year, long-term??
Author Response
Introduction
Pulpotomy includes partial pulpotomy, cervical pulpotomy full pulpotomy…etc. The word full pulpotomy is used in this manuscript. Add the explanation pulpotomy and define.
Answer: We complete the description as follows:
The success rate for this procedure varies widely from 56% to 100% [3–5]. The alternative approach is to treat pulpitis with a full pulpotomy (removal of the entire coronal pulp to the level of canal) [6–9] to restore coronal damage with direct plastic materials with cusp coverage [9] or preformed crowns [10].
“This study aimed at evaluating the outcome of the original combination of full Pulpotomy and Immediate CAD/CAM End-crown (PICCE) on teeth with large carious lesions and pulpitis”(Page 2, Line 67). Did the conclusion correspond to individual sentences?
Answer: Thank you for this remark, we changed as follow the conclusion:
Within the limitations of this preliminary study, full Pulpotomy and Immediate CAD/CAM End-crown (PICCE) could be safely recommended to treat, during a single session, severely damaged permanent molars with pulpitis. This new combination preserves the residual biological potential of the pulp, ensuring an endo-prosthetic continuum with the objective of long-term success.
The author investigated success rate of CAD/CAM endo crown without control. This should be stated as limitation.
Answer:
We agree with this point and we thank for this suggestion . We propose to include the following elements in the limits of the study section of discussion:
Material and methods
The author described “All pieces were polished before bonding. End-crowns were sealed with a dual cure resin cement” (Page 4, Line 116). Bonding procedure is important for CAD/CAM restoration. Did you perform sandblasting and priming??
Answer: indeed, our description was imprecise, we are modifying as follow:
“All pieces of IPS e.max® CAD were polished before bonding All endocrowns were etched with hydrofluoric acid during 20 seconds for IPS e.max® CAD and 30 seconds for Enamic® respectively. After priming, endocrowns were sealed with a dual cure resin cement (Variolink® Esthetic DC) (Figure 3). A postoperative periapical radiograph was taken after restoration placement using a film holder and a paralleling technique.”
Describe details such as clinical procedures Microscope video should be shown as new figure. Presenting these results proves your procedure.
Answers: We don't have the video module for the microscope. We can’t produce a video for the 23rd of august, that is the required date for the revised version of the manuscript.
In addition, the author described “A postoperative periapical radiograph was taken after restoration placement using a film holder and a paralleling technique.” (Page 4, Line, 118). Clinical date such as radiographs are not enough in this manuscript. Representative series of clinical images (X-ray photo, CBCT image) and videos (microscope), intraoral photo of each group should be open to help the reader. Especially, pre-ope, post-ope and follow-up, these time points is important.
Answer: We present step by step the all procedure with the addition of 3 news figures (Figures, 2,3,4) including photography, CadCam screenshots and X-ray images.
Remark: As American Association of Endodontists *, we didn’t realize a CBCT and couldn’t complete the illustration of the article for ethical reason (Descriptive study frame)
AAE and AAOMR Joint Position Statement: Use of Cone Beam Computed Tomography in Endodontics 2015 Update.
Special Committee to Revise the Joint AAE/AAOMR Position Statement on use of CBCT in Endodontics. Oral Surg Oral Med Oral Pathol Oral Radiol. 2015 Oct;120(4):508-12. doi: 10.1016/j.oooo.2015.07.033. Epub 2015 Aug 3. PMID: 26346911 Review.
Result
In figure1 End-crown and full pulpotomy, where is circular butt joint??
Answer: We agreed the remark. That was a mistake; we didn’t carry out a peripheral ceramic-tooth joint. It was remove of the section.
In addition, photograph of end-crown made using CAD/CAM is should be added as figures.
Answer: it was inserted in the manuscript.
“12 patients (40%) were lost to the study during follow-up.“(Page 8, Line163).
It seems high rate. Is this acceptable??
Additionally, how did you control the bias through this study?.
Answer: We fully agree with these limitations of the study; they have been clarified in the paragraph of the discussion on study limitations (see below)
Discussion
The author evaluated shor time out come of CAD/CAM endocrown, but did’t mention long trem follow-up. This part sould be stated as limitation of this paper.
Result of this study should be compared with previous study. Additionaly, this study was not randamized control trial. It is important to note limitation.
Answer:
We agree with this limits of the study, it was included in in the paragraph of the discussion on study limitations as follow
“This study presents some limitations especially related to the analysis strategy and high rate of subjects lost to follow up rate. Indeed, the data was not analyzed using an intention to treat approach. An analysis carried out whereby the denominator (n=30) stays the same, regardless of what happens to the patient, the assumption made is that all the patients who were lost to follow up had negative outcomes, the success rate would be significantly less favorable to the treatment procedure presented. As such, it is likely that the true success rate lies somewhere between the percentages given and the percentages obtained using an intention-to-treat analytical approach. The difficulty of strictly following the study protocol within the ethical setting and consequently the high loss to follow up rate, impacts this study results. In addition, the study design does not include a randomization procedure and a control group. Consequently, the results of this study need to be confirmed in a longer-term follow-up to allow comparison with data in the literature, as the indirect CAD/CAM endocrowns are expected to provide better long-term results than direct restorations in pulpotomised teeth.”
Conclusion
“This new combination preserves the residual biological potential of the pulp, ensuring an endo-prosthetic continuum with the objective of long-term success.” ((Page 8, Line 235). Follow-up period was one year, long-term?? and must be follow on a long term period
Answer: we qualify the conclusion as follow:
Within the limitations of this preliminary study, full Pulpotomy and Immediate CAD/CAM End-crown (PICCE) could be safely recommended to treat, during a single session, severely damaged permanent molars with pulpitis. This new combination preserves the residual biological potential of the pulp, ensuring an endo-prosthetic continuum with the objective of long-term success.
Round 2
Reviewer 1 Report
I am happy with the changes made although I feel my description of the study design is more specific than the broad term used by the authors (descriptive study!!).
Just one minor change noted in the manuscript and I have added a sticky note on the manuscript to reflect this.
It would be great if you could eventually do a trial on this topic. Well done though and I wish you all the best.

Author Response
We thank again Rewiever 1 for his/her boosts.
We agreed with replacing the term "descriptive study" with "cases reports on a series of patients". That was done on line 73.
We delete the word "rate" , line 199.
Reviewer 2 Report
The author has improved on the pointed out except for the provision of the video images.
Author Response
We thank Reviewer 2. We apologize for not being able to produce a video in the requested time for the edition process.